# The gender dimensions of mental health during the Covid-19 pandemic: A path analysis

**Kate Dotsikas**[1,2]*, **Liam Crosby**[1], **Anne McMunn**[3], **David Osborn**[1], **Kate Walters**[4], **Jennifer Dykxhoorn**[1,4]

**1** Division of Psychiatry, UCL, London, United Kingdom, **2** Institut Pierre Louis d'Épidémiologie et de Santé Publique, Equipe de Recherche en Epidémiologie Sociale, INSERM, Sorbonne Université, Paris, France, **3** Department of Epidemiology and Public Health, UCL, London, United Kingdom, **4** Department of Primary Care and Population Health, UCL, London, United Kingdom

* kge.dotsikas@gmail.com

**Data Availability Statement:** The data are publicly available upon creation of an account with the UK data service. The main UKHLS survey data is available at the following link: https://beta.

## Abstract

### Background

The Covid-19 pandemic has had a substantial population mental health impact, with evidence indicating that mental health has deteriorated in particular for women. This gender difference could be explained by the distinct experiences of women during the pandemic, including the burden of unpaid domestic labour, changes in economic activity, and experiences of loneliness. This study investigates potential mediators in the relationship between gender and mental health during the first wave of the Covid-19 pandemic in the UK.

### Methods

We used data from 9,351 participants of Understanding Society, a longitudinal household survey from the UK. We conducted a mediation analysis using structural equation modelling to estimate the role of four mediators, measured during the first lockdown in April 2020, in the relationship between gender and mental health in May and July 2020. Mental health was measured with the 12-item General Health Questionnaire (GHQ-12). Standardized coefficients for each path were obtained, as well as indirect effects for the role of employment disruption, hours spent on housework, hours spent on childcare, and loneliness.

### Results

In a model controlling for age, household income and pre-pandemic mental health, we found that gender was associated with all four mediators, but only loneliness was associated with mental health at both time points. The indirect effects showed strong evidence of partial mediation through loneliness for the relationship between gender and mental health problems; loneliness accounted for 83.9% of the total effect in May, and 76.1% in July. No evidence of mediation was found for housework, childcare, or employment disruption.

### Conclusion

The results suggest that the worse mental health found among women during the initial period of the Covid-19 pandemic is partly explained by women reporting more experiences

ukdataservice.ac.uk/datacatalogue/studies/study?id=6614 The Covid-19 survey is available at the following link: https://beta.ukdataservice.ac.uk/datacatalogue/studies/study?id=8644.

**Funding:** Kate Dotsikas, Liam Crosby, David Osborn, Kate Walters and Jennifer Dykxhoorn were supported by the National Institute of Health Research, School for Public Health Research (SPHR) Public Mental Health Programme. https://sphr.nihr.ac.uk/category/research/public-mental-health/ KD, DO, and JD were additionally supported by the NIHR University College London Hospital Biomedical Research Centre. https://www.uclhospitals.brc.nihr.ac.uk/our-research Anne McMunn is supported by the Joint Programming Initiative More Years Better Life from the following national funding body: UK Economic and Social Research Council (ES/W001454/1) and the UK ESRC International Centre for Lifecourse Studies in Society and Health (ICLS) (ES/J019119/1) https://jp-demographic.eu/ The funders had no role in study design, data collection and analysis, decision to publish, or preparation of the manuscript.

**Competing interests:** The authors have declared that no competing interests exist.

of loneliness. Understanding this mechanism is important for prioritising interventions to address gender-based inequities that have been exacerbated by the pandemic.

## Introduction

Covid-19 and the associated public health response have had significant social, economic and health impacts [1]. The initial public health policy response in the UK included a nation-wide lockdown, which involved restrictions on social contact, limits to economic activity, and school closures. This lockdown was implemented from March 23 until May 11, 2020, when some restrictions were relaxed; further restrictions were lifted in early July [2]. Substantial economic and social disruption occurred because of these measures [3,4] that may have implications for mental health.

Evidence has emerged demonstrating that mental health deteriorated substantially in the UK during the pandemic compared to pre-pandemic levels [5–10]. Prior to the Covid-19 pandemic, gender disparities in mental health and wellbeing were known to exist, with women exhibiting greater risk of common mental disorders, including depression and anxiety, than men [11]. This disparity has widened during the pandemic, with women experiencing a higher average increase in mental distress [7,12]. One study showed that women's mental health scores had deteriorated twice as much as had men's between the 2017–19 baseline and April 2020 [13].

There are several possible explanations for the observed gender differences in mental health during the initial Covid-19 lockdown, including unequal division of domestic labour, differential impact of economic disruption, and different experiences of loneliness. Understanding the relative importance of these pathways is important to prioritising intervention to address gender-based inequities.

First, the gender difference can be explained by the burden of unpaid domestic labour. Evidence indicates that gender divisions of household labour existed prior to the pandemic [14]. Among opposite-gender couples in the UK, women take on 60% more unpaid domestic work, including housework and childcare [15]. This burden of unpaid work has been associated with greater psychological distress [16]. The higher level of psychological distress may be driven by role overload, where an individual takes on multiple roles, such as household manager, primary caregiver, cook, and cleaner in addition to paid employment outside the home, which can deplete a person's limited energy capacity [16]. These gender inequalities have been shown to persist during the first Covid-19 lockdown: in opposite-gender households, women spent more time on household responsibilities, even among couples where both adults are engaged in paid work [17]. Such women reported higher levels of psychological distress for every one-hour increase in hours per week of housework and childcare during the first lockdown period [18].

The large-scale economic disruption caused by the pandemic may have had unequal gender impacts. Women were one-third more likely than men to work in sectors that were impacted by stay-at-home lockdown measures, including hospitality and retail [19], and were also more likely to quit, lose their job or be furloughed during the pandemic [17,20,21]. This disruption to the economic activity of many women in the UK during the initial lockdown may have led to worse mental health. A large body of evidence shows that unemployment carries an increased risk of poor mental health [22]. In the context of the Covid-19 pandemic, one UK study found that adults who became unemployed, were made redundant, or whose work hours were reduced during the first lockdown were over two times more likely to develop a

common mental disorder by July 2020, compared to self-employed adults whose business were not affected by the pandemic [9].

Finally, experiences of loneliness increased during the pandemic [23]. Loneliness reflects perceived social isolation, a subjective experience of a discrepancy between actual and desired social engagement [24,25]. In the UK, women were among those who were most at risk of loneliness during the first lockdown [26]. There is a strong evidence base for the association between loneliness and mental health problems [27,28]; evidence suggests that experiencing loneliness can increase the risk of the onset of depression [29].

The persistence of these changes is unknown: some UK evidence suggests overall distress, and the gender disparity therein, may have diminished in the months following the initial lockdown period [30,31], whereas literature reviews from other quarantines suggest potentially long-lasting negative psychological effects [32,33]. Specifically, the hypothesised pathways may be of different relative importance to mental health in the long-term compared to the short-term. For example, schools in England reopened on June 1, 2020, which may have led to a reduced role for the burden of childcare in the relationship between gender and mental health. Similarly, reductions in social contact had mostly ended by July 2020, potentially lessening the impact of loneliness. In contrast, the furlough scheme and the shutdown of certain sectors persisted for many months into the pandemic, suggesting that employment disruption may play a more important role in the long term.

This study aims to identify potential mediators in the relationship between gender and mental health during the first wave of the Covid-19 pandemic in the UK. We hypothesised that loneliness during the initial lockdown period would be the most important pathway for short-term mental health outcomes, while employment disruption may account for larger proportion of the gender difference in mental health outcome in the longer-term. We also hypothesised that while housework and childcare would not be as important as loneliness, they would play an approximately equal role across both time points.

## Methods

### Data

We used data from Understanding Society, the United Kingdom Household Longitudinal Survey (UKHLS). UKHLS is a nationally representative prospective panel survey that collects information annually on the health, wellbeing and socioeconomic situation of individuals and households in England, Scotland, Wales, and Northern Ireland [34]. All adult respondents to Waves 8 (2016–2018) or 9 (2017–2019) of the full survey were invited to participate in monthly Covid-19 surveys from April 2020 onwards; this initial wave had a 42% response rate [35]. After the July wave, those from the full UKHLS survey that were eligible for participation in the Covid-19 survey continued to be eligible in subsequent waves, irrespective of their participation in previous Covid-19 study waves. Baseline pre-pandemic information for our study was taken from interviews for the full UKHLS survey conducted during the 2019 calendar year, which come from a combination of Waves 10 (2018–2020) and 11 (2019–2021). Follow-up data was taken from the first four waves of the UKHLS Covid-19 study (April-July 2020) which cover the first period of lockdown in the UK.

No consent was obtained for this study, as it involved the secondary use of existing data, for which consent was already obtained by the Understanding Society research team. Data collection for the Understanding Society survey was approved by the University of Essex Ethics Committee, and oral consent was obtained at each wave [36].

## Participants

We developed a cohort of UKHLS participants who were full or partial respondents in the 2019 calendar year, and in April, May and July of the Covid-19 study waves, with at least one wave of complete mental health data. Of the 17,761 UKHLS participants that gave a response to the April Covid-19 study wave, 6,222 were excluded as they did not provide a response in 2019 or in the May and July Covid-19 waves. We excluded 1,348 participants that had a longitudinal weight of zero in Wave 9 of UKHLS and 479 who did not respond in both May and July Covid-19 waves. We further excluded those who did not provide GHQ-12 data at any of the included waves (n = 4), as well as 14 participants with no data on ethnicity or country, leaving a final analysis sample of 9,694.

## Measures

The main outcome for our study was the 12-item General Health Questionnaire (GHQ-12) [37], taken from May and July of the Covid-19 study. Respondents score between 0 and 4 on each item, producing a continuous Likert score between 0 and 36, where 0 indicates no mental health problems and 36 indicates high mental health problems.

The main exposure was gender. There is no gender information collected in UKHLS, therefore sex was used as a proxy. Sex is a binary variable with the values Male or Female and was measured in 2019.

We evaluated four potential mediators in the relationship between gender and mental health, all of which were measured at the April Covid-19 wave. Employment disruption was measured using a binary variable of whether participants had experienced a reduction in their working hours since January/February 2020. The burden of childcare and housework was measured with continuous variables that measured the number of hours spent per week on childcare and housework, respectively. The top 1% of respondents (reporting >90 weekly childcare hours) and the top 5% of housework (reporting >49 hours weekly on housework) were treated as erroneous and were recoded as missing. Finally, loneliness was measured with an ordinal variable that recorded the frequency of feelings of loneliness. Participants were asked "In the last 4 weeks, how often did you feel lonely?" with three response levels: 'hardly ever or never', 'some of the time', and 'often'. This measure of loneliness is taken from the Government Statistical Service harmonised principle of loneliness.

Covariates included age in 2019 (as a continuous variable), equivalised household income quintiles at Wave 10 (2018–2020) and GHQ score in 2019.

## Statistical analysis

**Preliminary analysis.** The sample was described in terms of the analysis variables. The difference in means between men and women were evaluated with unpaired t-tests (continuous variables), and the chi-square test was used for categorical variables.

We estimated the direct effect between gender and mental health problems in both May and July using unadjusted and adjusted linear regression models. We also examined the association between gender, the four mediators, and mental health problems in separate regression models at each time point.

**Path model for mediation.** We conducted a path analysis for the relationship between gender and mental health using structural equation modelling, testing the hypothesised causal pathway (Fig 1).

In this model, all variables are endogenous except for household income, age, and pre-pandemic mental health. We used Weighted least square mean and variance adjusted (WLSMV) estimation to account for the presence of categorical mediators. WLSMV uses linear regression

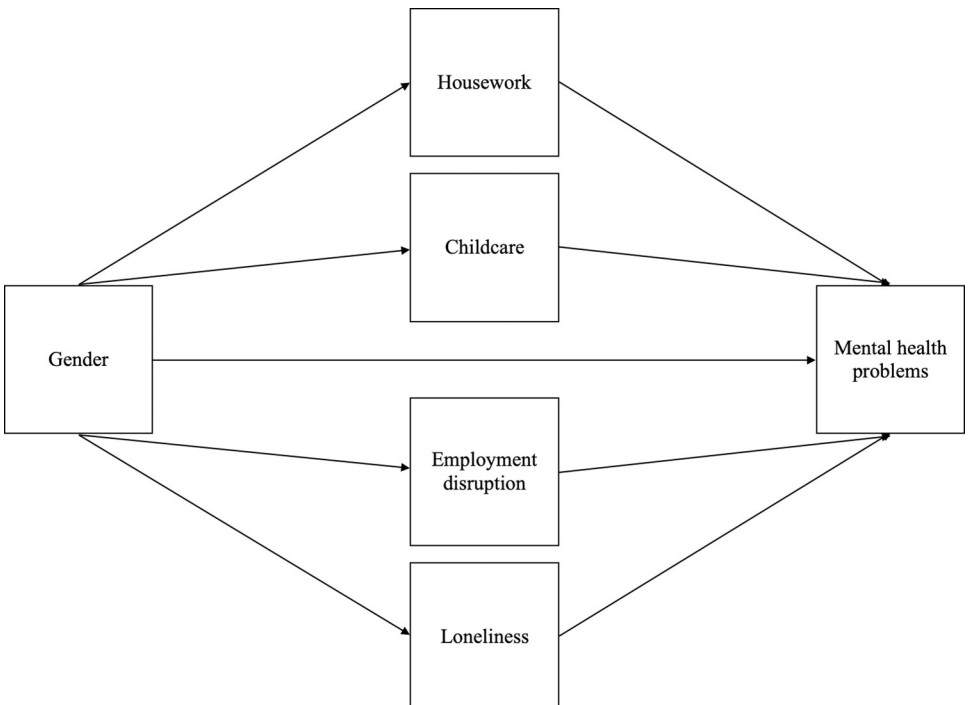

**Fig 1. Hypothesised model for the relationship between gender and mental health during the Covid-19 pandemic.**

to estimate the relationship between an independent variable and a continuous dependent variable (including mediators), and probit regression to estimate the relationship between an independent variable and a categorical dependent variable. Therefore, in this model, the latent response variables underlying the observed categorical variables are considered in the estimation, instead of the observed variables.

The coefficients for each relationship in the path diagram were obtained, as well as the indirect effects of the relationship between gender and mental health through each mediator pathway. We evaluated mediation as the proportion of the total effect mediated by each indirect effect. Indirect effects were calculated using the product of coefficients method. Coefficients were standardized to allow for comparison. We produced bootstrapped standard errors with 5000 draws to account for the non-normality of the distribution of indirect effects. We evaluated model fitness using the following fit indices: Root Mean Square Error of Approximation (RMSEA; good fit <0.06 and adequate fit <0.08), Comparative fit index (CFI; good fit >0.95 and adequate fit >0.9), and Tucker-Lewis index (TLI; good fit >0.95 and adequate fit >0.9) [38].

**Survey design and probability weights.** Due to the complex survey design of UKHLS, all stages of the analysis took into account stratification and the primary sampling unit. Furthermore, probability weights were included in the analysis to account for sampling probability and bias due to nonresponse. We used customised weights, following the procedure described in Benzeval et al. [39]. Specifically, the weights were designed to account for response to the UKHLS waves of interest for this project. Weight production was contingent on being a respondent to Wave 9 of the UKHLS main survey, a respondent in 2019, and a respondent to the April 2020 Covid-19 wave. Additionally, weights were restricted to those responding in each of the two outcome waves, resulting in two weights being produced: one for the model using mental health score in May, and one for the model using July mental health score.

**Missing data.** WLSMV uses pair-wise deletion [40]; thus, we excluded participants who had missing data on exogenous variables: age, household income, and pre-pandemic mental health.

Data management took place in Stata 17. The path analysis was estimated with MPlus 8.

## Results

### Sample

Following exclusions, our analytic sample was 9,351, including those who had participated in 2019 and the Covid-19 waves (April and May/June) and had complete data on exogenous variables.

The distribution of sample characteristics can be found in Table 1. The final sample was made up of slightly more men than women (58.2% vs 41.8%), with a total mean age of approximately 55. The mean mental health problems score for the total sample was at its lowest pre-pandemic, and at its highest in May 2020, with the mean score exceeding the cut-off that is consistent with a diagnosis of a common mental disorder. Across all three time points, women had a higher mean mental health problems score than men, with the greatest difference seen in May (12.7 vs 11.04). Compared to men, a greater proportion of women reported feeling lonely some of the time (31.5% vs 20.5%) or often (7.9% vs 4.1%), and women on average spent more time on childcare (4.5h/day vs 2.3h/day) and housework (14.5h/day vs 9.6h/day).

### Missingness

The proportion of missing data for each variable can be found in S1 Table. Of the variables with missing data, there was a low level of missingness (under 3% for each variable). Hours of housework had the greatest proportion of missing data. In the case of this study, 3.5% (n = 343) of the sample has missing data on the exogenous variables, age, household income, and pre-pandemic

**Table 1. Descriptive characteristics.**

| Variable | Women (n = 5,439) | Men (n = 3,912) | Group Differences |
|---|---|---|---|
| Age–mean (SD) | 53.3 (15.5) | 56.6 (15.2) | $t(9349) = 10.23$* |
| GHQ (Pre-pandemic)–mean (SD) | 11.6 (5.5) | 10.3 (4.9) | $t(9349) = -11.27$* |
| GHQ (May)–mean (SD) | 12.7 (5.9) | 11.04 (5.3) | $t(9274) = -14.08$* |
| GHQ (July)–mean (SD) | 11.9 (5.6) | 10.7 (5.1) | $t(9227) = -11.24$* |
| Loneliness–n (%) | | | $x^2(2) = 227.08$* |
| Hardly ever or never | 3,290 (60.6) | 2,941 (75.3) | |
| Some of the time | 1,711 (31.5) | 802 (20.5) | |
| Often | 430 (7.9) | 161 (4.1) | |
| Hours of childcare–mean (SD) | 4.5 (13.4) | 2.3 (8.8) | $t(9211) = -7.86$* |
| Hours of housework–mean (SD) | 14.5 (8.8) | 9.6 (7.6) | $t(9084) = -27.6$* |
| Employment Disrupted–n (%) | 1,210 (22.3) | 844 (21.7) | $x^2(1) = 0.57$ |
| Household Income Quintiles–mean (SD) | | | |
| Very High | 4,039.7 (1700.6) | 4,080 (1686.9) | $t(1873) = 0.52$ |
| High | 2,474.8 (200.2) | 2,483.8 (196.6) | $t(1870) = 0.97$ |
| Middle | 1,924.9 (137.6) | 1,926.2 (135.5) | $t(1870) = 0.21$ |
| Low | 1,489.9 (119.6) | 1,493.9 (121.2) | $t(1871) = 0.69$ |
| Very Low | 925.4 (306.2) | 903.6 (327.4) | $t(1857) = -1.45$ |

*pr (|T| > |t|) <0.001 *p<0.001.

mental health problems score. The estimates produced by WLSMV can be biased in the presence of data that may be missing at random with respect to both covariate variables and observed dependent variables (MAR). However, this bias was assumed to be negligible with the low level of missing data present in the sample. Moreover, no significant differences were found between the characteristics of those included in the analysis and those that were dropped. The comparison between included and excluded cases can be found in S2 Table.

## Preliminary analysis

The results of the regression models testing the direct effect between gender and GHQ in May and July can be found in S3 Table. After controlling for age, household income and 2019 GHQ, we found evidence of a direct relationship between gender and GHQ in both May and July.

For the preliminary analysis, only cases with complete data on sex, age, household income, and all three GHQ time points were included. Therefore, in the linear regression model, a further 1.9% (n = 181) of the sample was dropped.

## Path analysis

Fig 2 displays the path diagram for the hypothesised mediation model for gender and mental health in May (a) and July (b), with standardised coefficients and bootstrap standard errors for

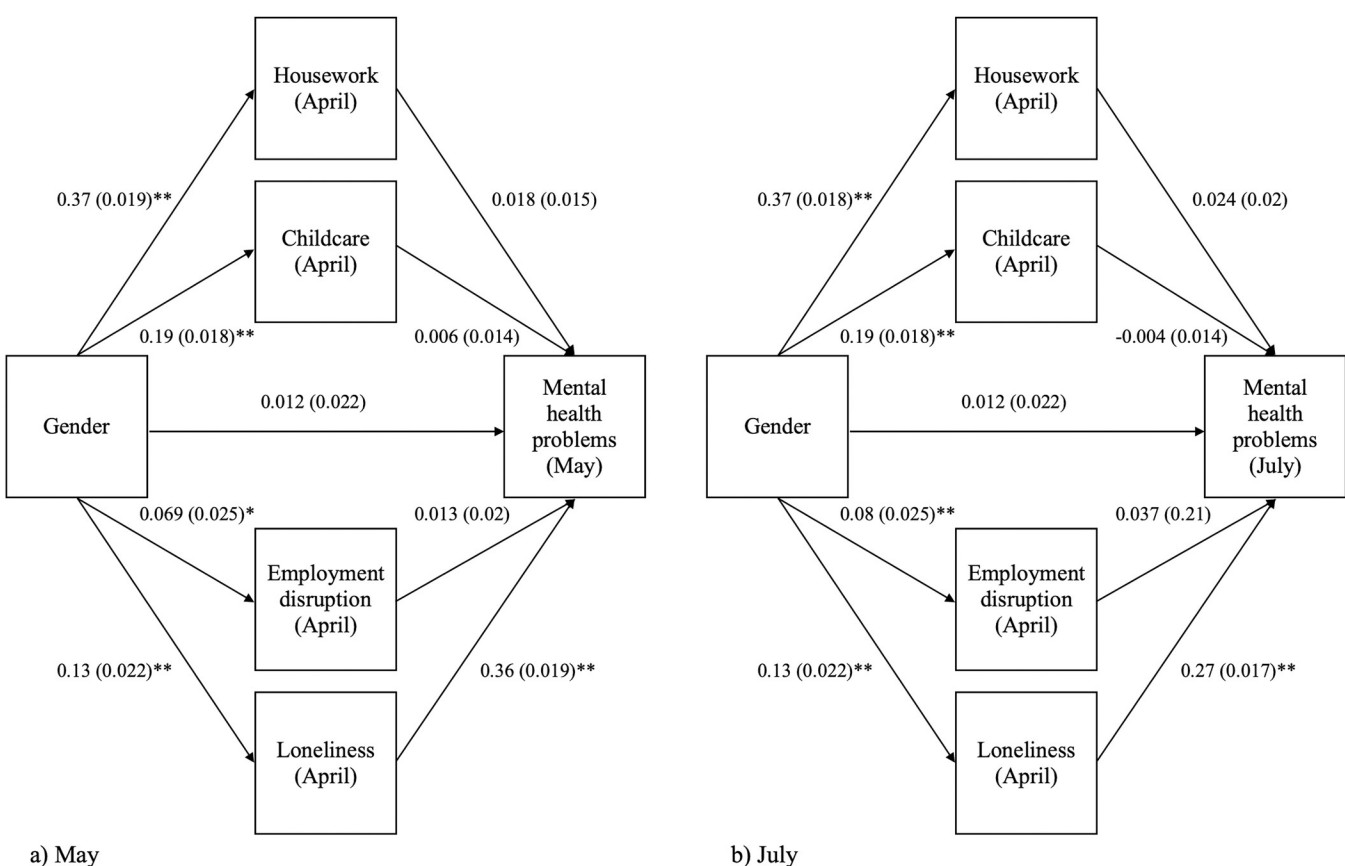

**Fig 2. Path analysis for the relationship between gender and mental health in May 2020 (Model 1) and June 2020 (Model 2).** Adjusted for age, household income quintile, pre-pandemic mental health score *p<0.05 **p<0.001.

each path. For both models, the value of RMSEA met the criteria for good fit and the CFI indicated adequate fit, however the TLI value was under the threshold for adequate fit. The values for each fit indices can be found in S4 Table.

## Path estimates

Out of the four mediators, gender most strongly predicted housework in both models, with women being more likely to spend longer on housework (Fig 2). Gender also predicted the remaining mediators, with the effect size for the relationship between gender and employment disruption increasing slightly in July. No evidence of a direct effect of gender on mental health was found in the full path models.

Loneliness, measured in April, was strongly associated with mental health problems in May and June (Fig 2). Higher loneliness scores in April 2020 were associated with more mental health problems in May and June. The evidence did not support an association between the remaining mediators and mental health in May or July.

## Mediation

When exploring the indirect effects (Table 2), we found evidence that the relationship between gender and mental health problems was partially mediated through loneliness. In the first model, loneliness accounted for loneliness accounted for 83.9% of the total effect, and 76.1% of the association between gender and mental health problems was mediated through loneliness in the second model. No evidence of mediation was found for housework, childcare, or employment disruption.

## Discussion

In this nationally representative panel survey, loneliness mediated the relationship between gender and mental health problems during the first wave of the Covid-19 pandemic.

After controlling for pre-pandemic mental health, we showed that the worse mental health reported by women in the initial period of Covid-19 was partially explained by experiences of loneliness. Experiences of loneliness, measured in April, had a strong impact on mental health in May, and continued to have an impact on mental health problems in June, but to a lesser extent, suggesting that the mental health impact of loneliness has a persistent impact on the mental health of women. We found no mediating role for housework, childcare, or

**Table 2. Standardised indirect effects with bootstrapped standard errors.**

| May | Standardised coefficient | S.E. | 95% CI | P-value |
|---|---|---|---|---|
| Total effect | 0.056 | 0.11 | 0.034, 0.078 | <0.001 |
| Loneliness | 0.047 | 0.008 | 0.032, 0.063 | <0.001 |
| Housework | 0.007 | 0.006 | -0.004, 0.018 | 0.25 |
| Childcare | 0.001 | 0.003 | -0.004, 0.007 | 0.69 |
| Employment disruption | 0.001 | 0.002 | -0.002, 0.004 | 0.55 |
| **July** | **Standardised coefficient** | **S.E.** | **95% CI** | **P-value** |
| Total effect | 0.046 | 0.011 | 0.025, 0.068 | <0.001 |
| Loneliness | 0.035 | 0.007 | 0.023, 0.049 | <0.001 |
| Housework | 0.009 | 0.008 | -0.005, 0.023 | 0.24 |
| Childcare | -0.001 | 0.003 | -0.006, 0.005 | 0.78 |
| Employment disruption | 0.003 | 0.002 | 0.000, 0.007 | 0.12 |

employment disruption. We also did not find evidence of a direct effect between gender and mental health problems once all the mediators were taken into account.

The literature on loneliness and mental health among women is consistent with our findings. Prior to the Covid-19 pandemic, evidence of the relationship between gender and loneliness varied with other individual characteristics such as age, with a recent meta-analysis finding no significant gender difference in loneliness across the lifespan [41]. However, results from the pandemic reveal a larger prevalence of loneliness among women [42,43], in particular young women [44,45]. Longitudinal studies found a growing gender gap when measuring loneliness over time before and during the pandemic [26,46,47].

A large body of evidence shows that loneliness is associated with worse mental health, including depression [29,48–50]. While overall mental health has been shown to have improved by July 2020 [9,51,52], we found that loneliness in April 2020 continued to explain the gender gap in July 2020. Loneliness has been shown to have both short- and long-term effects on physical and mental health, which may explain this finding [53]. Loneliness can impact mental health by triggering hypervigilance for social threat, leading lonely people to perceive the social world as a threatening and negative place; thus, loneliness subsequently leads to feelings of hostility, stress and other persistent emotional states that present risk factors for poor mental health [24]. Women may be more vulnerable to the effects of loneliness on mental health: one study found that only objective social isolation, measured by frequency of social contacts, predicted depressive symptoms at follow-up among men, whereas both loneliness and social isolation predicted symptoms among women [54]. However, it is important to recognise that loneliness may be underestimated in men, who may be less likely to label themselves as lonely due to gender expectations [55–57].

Women in were more likely to be put on long-term furlough [58], giving them more time for housework and childcare, which may explain the absence of a mediating role for housework and childcare. Gender ideology may also explain this finding, as the impact of household labour on mental health may be different depending on whether women endorse traditional gender roles [59]; while women may spend more time on housework, they do not necessarily perceive this inequality as unfair [60], and therefore might not experience worse mental health. Regarding childcare, this analysis is based on a sample of adults of various household compositions, many of which do not include any children. Moreover, evidence for a mental health impact of childcare during the Covid-19 pandemic has been primarily found in lone parents [19]. Future research is needed to elucidate the potential mediator relationships for subgroups of lone parents and different gender ideologies.

No mediating role was found for employment disruption, which may be explained by the choice of measure. We measured employment disruption by a decrease in work hours since the start of the pandemic; individuals could report a decrease for a variety of reasons, including being furloughed, laid off, or having hours cut by an employer, and these reasons may present distinct relationships with gender and mental health. For example, while women were more likely to be working in sectors where the furlough scheme was heavily used [59], furlough may not have as severe a mental health impact compared to job loss [61]. However, women who had been on the furlough scheme were more likely to report perceived job insecurity [58]; evidence indicates that perceived job insecurity poses a comparable, and in some cases more severe, threat to mental health and wellbeing, compared to objective employment disruptions [62–65]. Therefore, the hypothesised causal model may be better specified using job insecurity.

To our knowledge, this study is the first to conduct a full mediation analysis to test the hypothesised causal relationship between gender and mental health in the Covid-19 context. The study benefits from the use of a large longitudinal sample representative of the UK

population, which allowed us to compare mental health before and during the pandemic. Furthermore, as the hypothesised pathway was tested using the outcome at two different time points, the results provide evidence of how experiences in April 2020 impacted the relationship between gender and mental health in May 2020, and how this impact persisted in July 2020 when lockdown measures were removed. Finally, the potential causal implications of the results are strengthened by a temporal separation between the exposure, mediators, and outcome.

Our study was limited by the use of sex as a proxy for gender, two related but distinct concepts. This study is concerned with the complex social phenomenon of gender as it is investigating differing behaviours, activities, and experiences between men and women, rather than differences in biological characteristics [66]. Unfortunately, no gender measure was available for UKHLS. Furthermore, loneliness has been shown to present a bidirectional relationship with mental health [67]. However, as included GHQ at baseline in our model, we adjusted for pre-existing mental health problems. Furthermore, GHQ was measured at a later time point than loneliness, ensuring temporal separation between these two variables to establish a directional relationship.

Finally, this study had a relatively short follow-up time. While we examined two time points for follow-up that represented different contexts as public health measures changed, they were only two months apart, and both close to the beginning of the Covid-19 pandemic. Evidence has emerged showing that rates of poor mental health fluctuated at different periods in the ongoing pandemic, and the gender gap correspondingly persisted [51,68]. Future research is needed to examine if the same mediating relationships persist at later time points.

In conclusion, results suggest that the worse mental health found among women during the initial period of the Covid-19 pandemic is partly explained by women reporting more experiences of loneliness. Understanding this mechanism is important for prioritising interventions to address gender-based inequities that have been exacerbated by the pandemic.

## Supporting information

**S1 Table. Missing data on analysis variables.**
(DOCX)

**S2 Table. Distribution of sample characteristics among complete cases and cases with missingness on exogenous variables.**
(DOCX)

**S3 Table. Linear regression model results for the relationship between gender and GHQ in May and July.**
(DOCX)

**S4 Table. Fit indices for the mediation models for gender and mental health in May and July.**
(DOCX)

## Acknowledgments

Thank you to Dr. Jamie C. Moore at the Institute for Social and Economic Research, University of Essex, for his help with the weight customisation.

## Author Contributions

**Conceptualization:** Kate Dotsikas, Anne McMunn, David Osborn, Kate Walters, Jennifer Dykxhoorn.

**Data curation:** Kate Dotsikas, Liam Crosby.

**Formal analysis:** Kate Dotsikas, Liam Crosby.

**Funding acquisition:** Kate Dotsikas, Liam Crosby.

**Investigation:** Kate Dotsikas, Liam Crosby.

**Methodology:** Kate Dotsikas, Anne McMunn, David Osborn, Kate Walters, Jennifer Dykxhoorn.

**Project administration:** Kate Dotsikas, David Osborn, Kate Walters, Jennifer Dykxhoorn.

**Supervision:** Anne McMunn, David Osborn, Kate Walters, Jennifer Dykxhoorn.

**Writing – original draft:** Kate Dotsikas, Liam Crosby.

**Writing – review & editing:** Kate Dotsikas, Anne McMunn, Jennifer Dykxhoorn.

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
