## [Decision Letter · Decision Letter 0]

9 Mar 2023

Explaining the gender gap in mental health during the Covid-19 pandemic: a path analysis using structural equation modeling

PONE-D-22-22332

Dear Dr. Dotsikas,

We’re pleased to inform you that your manuscript has been judged scientifically suitable for publication and will be formally accepted for publication once it meets all outstanding technical requirements.

Kind regards,

Prabhat Mittal, Ph.D.

Academic Editor

PLOS ONE

1. Please ensure that you have specified (1) whether consent was informed and (2) what type you obtained (for instance, written or verbal, and if verbal, how it was documented and witnessed). If your study included minors, state whether you obtained consent from parents or guardians. If the need for consent was waived by the ethics committee, please include this information.

2. Please amend either the title on the online submission form (via Edit Submission) or the title in the manuscript so that they are identical

Additional Editor Comments (optional):

The topic covered by the authors of this paper is unquestionably important and is consistent with data on global mental health disturbance. The reasoning in the study is supported by sound theoretical precepts. Path analysis with mediation have been carried out and completes all its assumption. The manuscript can be accepted in the present form. 

Reviewers' comments:

Reviewer's Responses to Questions

**Comments to the Author**

1. Is the manuscript technically sound, and do the data support the conclusions?

Reviewer #1: Yes

Reviewer #2: Yes

2. Has the statistical analysis been performed appropriately and rigorously? 

Reviewer #1: Yes

Reviewer #2: Yes

3. Have the authors made all data underlying the findings in their manuscript fully available?

Reviewer #1: Yes

Reviewer #2: Yes

4. Is the manuscript presented in an intelligible fashion and written in standard English?

Reviewer #1: Yes

Reviewer #2: Yes

5. Review Comments to the Author

Reviewer #1: The reviewed article explains the gender gap in mental health during the Covid-19 pandemic. The subject matter discussed by the article's Authors is undoubtedly essential and corresponds with mental health disruption worldwide statistics. The paper's argument is built on a proper theoretical basis and concepts. In the Methods section, the authors described exhaustively and in detail the materials and methods technique, research group, instrument, and analysis process. The statistical analysis is performed appropriately and rigorously. The obtained research results were presented clearly and compared with data from the literature. The conclusions correspond with the whole paper - refer to theoretical bases and research findings. Authors justify the need for intervention to address gender-based mental health inequalities exacerbated by the coronavirus pandemic. The literature cited in the article is adequate for the subject. The bibliography contains 67 items. The selection of the literature proves good theoretical preparation for the study. The paper identifies scientific and practical implications. The manuscript is presented in an intelligible fashion and written in standard English. Minor typing errors require corrections. In conclusion, I recommend the reviewed article for publication.

Reviewer #2: the study investigates potential mediators in the relationship

between gender and mental health during the first wave of the Covid-19 pandemic in the UK . General Health Questionnaire has been used to collect the data. the study evaluated four potential mediators in the relationship between gender and mental health, all of which were measured at the April Covid-19 wave. Hence the manuscript is technically sound and authenticated data sources has been used in the study. Findings are also inline with the analysis. Social implications are also clearly mentioned in the article.

6. PLOS authors have the option to publish the peer review history of their article (what does this mean?). If published, this will include your full peer review and any attached files.

Reviewer #1: No

Reviewer #2: **Yes: **Dr.G.Kavitha

---

## [Editor Report · Acceptance letter]

22 Mar 2023

PONE-D-22-22332 

The gender dimensions of mental health during the Covid-19 pandemic: a path analysis 

Dear Dr. Dotsikas:

I'm pleased to inform you that your manuscript has been deemed suitable for publication in PLOS ONE. Congratulations! Your manuscript is now with our production department. 

Kind regards, 

on behalf of

Dr. Prabhat Mittal 

Academic Editor

PLOS ONE